# 2-Phenylimidazole Corrosion Inhibitor on Copper: An XPS and ToF-SIMS Surface Analytical Study

Matjaž Finšgar

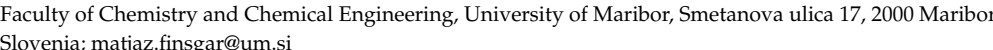

Faculty of Chemistry and Chemical Engineering, University of Maribor, Smetanova ulica 17, 2000 Maribor, Slovenia; matjaz.finsgar@um.si

**Abstract:** This work presents a surface analytical study of the corrosion inhibitor 2-phenylimidazole (2PhI) adsorbed on a Cu surface from 3 wt.% NaCl solution. X-ray photoelectron spectroscopy (XPS) and time-of-flight secondary ion mass spectrometry (ToF-SIMS) were used to investigate the surface phenomena. Various XPS experiments were performed, i.e., survey- and angle-resolved high-resolution XPS spectra measurements, gas cluster ion beam sputtering in conjunction with XPS measurements, and XPS imaging in conjunction with principal component analysis. These measurements were used to detail the composition of the surface layer at depth. In addition, various ToF-SIMS experiments were performed, such as positive ion ToF-SIMS spectral measurements, ToF-SIMS imaging, and cooling/heating in conjunction with ToF-SIMS measurements. This study shows that organometallic complexes were formed between 2PhI molecules and Cu ions, that the surface layer contained entrapped NaCl, that the surface layer contained some Cu(II) species (but the majority of species were Cu(I)-containing species), that the surface was almost completely covered with a combination of 2PhI molecules and organometallic complex, and that the temperature stability of these species increases when 2PhI is included in the organometallic complex.

**Keywords:** 2-phenylimidazole; corrosion inhibitor; ToF-SIMS; copper; XPS; GCIB





## 1. Introduction

The use of a corrosion inhibitor is a very convenient method to mitigate corrosion, especially for closed systems. Azoles are one of the most effective classes of organic corrosion inhibitors for various metallic materials [1–14]. Azoles contain nitrogen, which can serve as an active site for adsorption on the metal surface [15–17]. Recently, azoles have been shown to effectively mitigate galvanic corrosion [18], effectively inhibit multilayer copper wiring in integrated circuits [19] and are effective in various corrosive environments [20]. Depending on the type of azole used, these compounds can inhibit copper corrosion through various oxidation mechanisms [21].

The manner in which corrosion inhibitors bond to the surface is critical in determining whether they can effectively mitigate corrosion. The latter can be investigated using surface analytical techniques, in particular X-ray photoelectron spectroscopy (XPS) and time-of-flight secondary ion mass spectrometry (ToF-SIMS) [22–28]. The recent development of gas cluster ion beam (GCIB) technology in combination with surface analytical techniques enables a detailed study of very thin surface layers [29–35], e.g., the very thin surface layers of corrosion inhibitors. A GCIB source is especially beneficial in combination with XPS and ToF-SIMS [34]. A GCIB source is particularly useful for studying organic–inorganic interfaces where monoatomic Ar$^+$ sputtering (a previously common technology) would result in undesirable chemical changes that could lead to a misinterpretation of the interfacial structure, which in most cases is an essential property in the development of an effective corrosion inhibitor/metal material/corrosive solution system.

This work presents a surface analytical study of 2-phenylimidazole (2PhI, Figure 1a) adsorbed on a Cu surface from 3 wt.% NaCl solution. Using various electrochemical

techniques, 2PhI was previously shown to effectively protect Cu in chloride solution at a concentration of 1 mM [7]. However, no detailed surface analytical study has been carried out to explain why this compound acts as an effective corrosion inhibitor. Therefore, the manner of 2PhI bonding on Cu was investigated to explain the mechanism of surface adsorption and the formation of the protective surface layer, which is crucial for a corrosion inhibitory effect [36–38]. The adsorption of 2PhI was carried out in 3 wt.% NaCl solution, which is a very corrosive solution for Cu and therefore a suitable medium for studying corrosion inhibition phenomena. The nature of the surface bonding was investigated by means of XPS, in particular, by angle-resolved XPS (ARXPS) measurements and GCIB sputtering in conjunction with XPS measurements. In order to quantitatively identify the atomic composition of different species on the surface and to determine the average surface concentration in different phases, XPS imaging was performed in conjunction with principal component analysis (PCA). Subsequently, molecular-specific signals were measured by time-of-flight secondary ion mass spectrometry (ToF-SIMS) to provide additional information on the formation of organometallic complexes, to show the distribution of different species on the surface, and to reveal the temperature-dependent persistence of these species.

## 2. Experimental

### 2.1. Sample and Solution Preparation

Cu with a purity of 99.999% was supplied by Goodfellow (Cambridge, UK) and cut in the form of discs. A grinding device was used to grind the samples under a water flow starting with 500-grit SiC papers (Struers, Ballerup, Denmark), followed by 800-, 1000-, 1200-, 2400-, and 4000-grit SiC papers. Grinding with each SiC paper was performed by rotating the sample by 90°. After grinding, the samples were rinsed with ultrapure water with a resistivity of 18.2 MΩ cm. After rinsing, the samples were cleaned in an ultrasound bath for 3 min. The solution in the ultrasound bath contained 50 vol.% pure ethanol (absolute for analysis EMSURE® ACS, ISO, Reag. Ph Eur, Merck KGaA, Darmstadt, Germany) and 50% vol. (volume) ultrapure water. Subsequently, the samples were thoroughly rinsed with ultrapure water and dried under a flow of Ar gas [39].

The adsorption of 2PhI was carried out in 3 wt.% NaCl solution at 25 °C. 2PhI was supplied by Merck KGaA (with a purity of 98%). NaCl (for analysis EMSURE® ACS, ISO, Reag. Ph Eur) was supplied by Merck KGaA, Darmstadt, Germany. After sample immersion for 1 h, the samples were removed from the solution, rinsed with ultrapure water, dried under an Ar flow, and placed in an XPS or SIMS instrument. When the sample was prepared as described, the term 2PhI-treated Cu is used.

Both XPS and ToF-SIMS analyses were performed on three separately prepared samples, and the analyzed spots showed a high degree of spectral similarity. On this basis, the reported measurements are considered representative (measurements from one of the three analyzed spots are shown herein).

### 2.2. XPS Measurements

XPS measurements were performed using the Escalab Xi+ instrument (Thermo Fisher Scientific, East Grinstead, UK) controlled by Thermo Avantage software (Thermo Fisher Scientific, East Grinstead, UK). The latter software was also used for data processing after the measurements. To obtain XPS spectra, an Al $K_\alpha$ X-ray source was employed. Different pass energies were used to measure high-resolution (HR) and survey spectra, i.e., a pass energy of 200.0 eV for the former and 50.0 eV for the latter. The MAGCIS module was used to generate a GCIB at different accelerated voltages and cluster sizes. Sputtering was performed on a raster size of 400 by 400 μm. The subscript X in $Ar_x^+$ clusters in the designation used hereinafter represents the number of Ar atoms in the cluster. The binding energy ($E_B$) scale of the XPS spectra was corrected by the C–C/C–H peak in the C 1s peak, which is located at 284.8 eV. ARXPS measurements were performed at six different take-off angles (θ), while depth profiling was performed at θ = 90°. The θ is the angle with respect to

the surface and the path of the emitting photoelectrons. A Shirley background subtraction was performed before calculating the atomic concentrations (which were normalized to 100.0%). All HR XPS spectra presented in this work were subjected to Shirley background subtraction.

### 2.3. ToF-SIMS Measurements

ToF-SIMS measurements were performed with an M6 device (IONTOF, Münster, Germany) equipped with a Nanoprobe 50 bismuth cluster ion source. The measurements were controlled, and the data were processed with SurfaceLab 7.1 software (IONTOF, Münster, Germany). The device has a mass resolution of more than 30,000. The following signals were used to calibrate the spectra at different mass-to-charge ratios ($m/z$): $CH_3^+$ ($m/z$ 15.02), $C_2H_3^+$ ($m/z$ 27.02), and $C_3H_5^+$ ($m/z$ 41.04). The peak assignment was performed based on the $\Delta$ parameter (in ppm). The current for analysis was 0.3 pA, and the acquisition time to obtain spectra was at least 80 s. A 0.20 pA current for analysis was employed for ToF-SIMS imaging. The latter analysis was carried out on a 500 by 500 μm spot size.

## 3. Results and Discussion

In order to adsorb 2PhI molecules on the surface for the XPS and ToF-SIMS analyses, the Cu samples were immersed for 1 h in 3 wt.% NaCl solution containing 1 mM of 2PhI. A concentration of 1 mM is typical in corrosion inhibitor research and therefore also enables direct comparison with previous studies. In the discussion that follows, the XPS results are presented first, followed by the ToF-SIMS analyses in order to describe the surface bonding and distribution of 2PhI on the Cu surface.

XPS and ToF-SIMS measurements were also performed for the Cu sample before immersion (after the preparation procedure as described in Section 3.1) and for the sample that had been immersed in 3 wt.% NaCl without 2PhI. XPS analysis of the sample after the preparation procedure showed the signals for Cu- and O-containing species, which represent an oxidized Cu surface. Moreover, the C 1s XPS signal was intensive, which corresponds to the adsorption of adventitious carbonaceous species. The same results were obtained using ToF-SIMS, which detected the signals characteristic of the oxidized Cu surface containing typical hydrocarbon fragments. Moreover, the surface of the sample that had been immersed in 3 wt.% NaCl solution also contained the same species as the sample after the preparation procedure. On the other hand, no chlorides were detected on the surface—no signal for Cl 2p was detected (when using XPS), and no signal at $m/z$ 34.67 in negative polarity, corresponding to chloride, was detected (when using the ToF-SIMS technique).

### 3.1. Elemental-Specific Analysis Using XPS

Figure 1 shows the survey and HR ARXPS spectra for 2PhI-treated Cu. As the substrate was made of pure Cu, several peaks originating from it appeared in the survey spectrum (Figure 1b), i.e., Cu 3p, Cu 3s, Cu 2p, and XPS-excited Auger peaks (Cu $L_3M_{4,5}M_{4,5}$, Cu $L_3M_{2,3}M_{4,5}$, and Cu $L_3M_{2,3}M_{2,3}$). As explained below, Cu-related signals also originate from the organometallic complexes that formed between 2PhI molecules and Cu ions. Cu ions were released during the initial stage of Cu corrosion in 3 wt.% NaCl solution [40]. The intensive O 1s peak mostly originates from oxidized Cu (oxides and/or hydroxides). Moreover, the O 1s peak can also originate from oxidized adventitious carbonaceous species that adsorbed on the surface after the sample preparation procedure and its transfer to the spectrometer. The contribution to the O 1s peak intensity due to water molecules that remained on the surface and may be hydrogen-bonded to the 2PhI surface layer cannot be excluded.

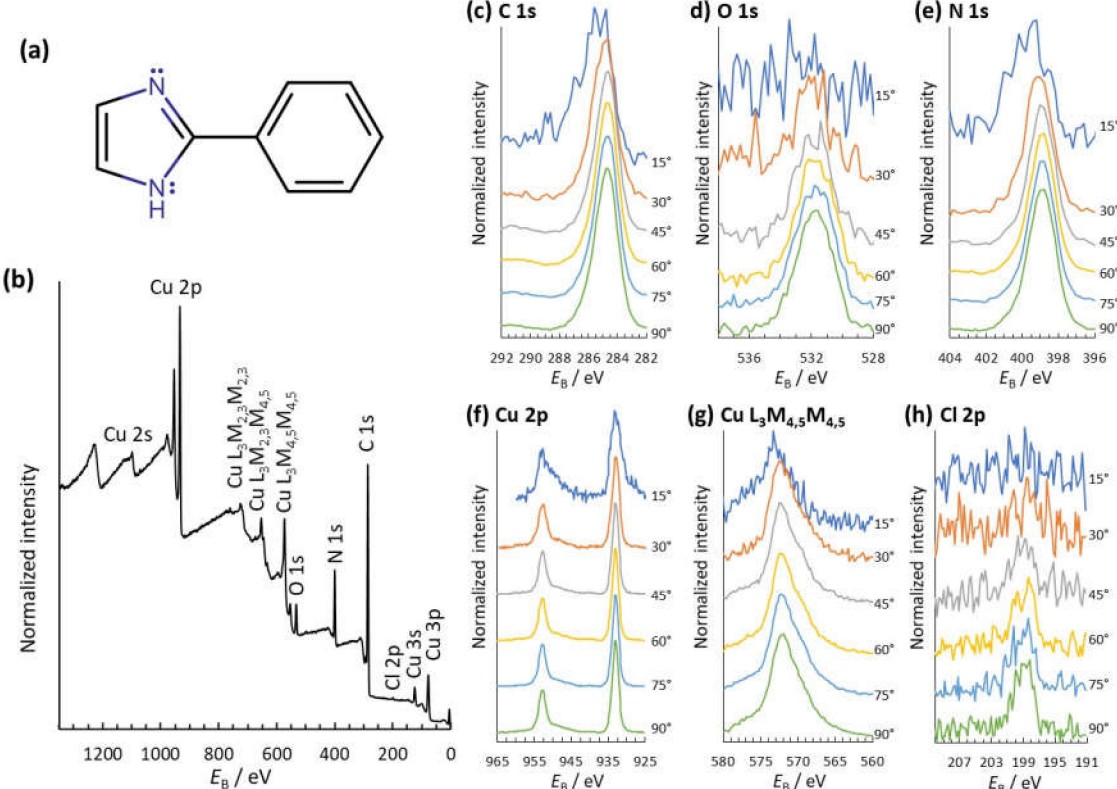

**Figure 1.** (**a**) The chemical structure of the 2PhI molecule. XPS analyses of the 2PhI-treated Cu: (**b**) survey and HR ARXPS spectra for (**c**) C 1s, (**d**) O 1s, (**e**) N 1s, (**f**) Cu 2p, (**g**) XPS-excited Cu $L_3M_{4,5}M_{4,5}$, and (**h**) Cl 2p.

A clear indication that 2PhI molecules were adsorbed on the Cu surface is the presence of an N 1s peak (no other species possibly adsorbed on the surface contained N atoms). The adsorption of 2PhI is also supported by the presence of a C 1s peak (there are C atoms in the 2PhI structure). However, the C 1s signal could also originate from adventitious carbonaceous species. A non-intensive peak at an $E_B$ of approximately 199 eV corresponds to Cl 2p (Figure 1b), which originates from NaCl, where chlorides were probably entrapped inside the surface layer [41–43]. Moreover, an Na$^+$ signal was detected using the ToF-SIMS technique (as explained below), thus confirming NaCl on the surface. An intensive background contribution in an $E_B$ range more positive than approximately 560 eV in the survey spectrum (Figure 1b) suggests a relatively thick surface layer and implies the need for sputtering sources (as explained below).

Next, angle-resolved XPS analyses were performed at six different θ, i.e., 15°, 30°, 45°, 60°, 75°, and 90°. With a decrease in θ, the surface sensitivity increases and only the signal from the topmost position is obtained. By increasing the θ, the signal comes from the topmost species along with the species located deeper in the subsurface region. With an increase in θ, the position of the C 1s spectrum does not change significantly (Figure 1c). On the other hand, the level of spectral noise becomes lower. In the case of the O 1s and Cl 2p spectra, the central peak was not expressed at the lowest θ (Figure 1d,h), suggesting that the topmost position is not rich in O- and Cl-containing species. With an increase in θ, the peak in the O 1s and Cl 2p spectra becomes intensive, and the degree of noise becomes lower. The position of the central peak in the O 1s spectra is located at an $E_B$ of 532 eV, which suggests the presence of oxidized organic species. In the case of the N 1s spectra, the central peak is transferred to more negative $E_B$ with an increase in the θ from 15° to 30°. This change implies two different N environments. The N environment of the species in the topmost position most likely corresponds to the organometallic complex that was formed between the 2PhI molecules and Cu ions, which were released during the initial stage of Cu corrosion. The 2PhI molecules are most likely located underneath the organometallic

complex (not in the organometallic complex). With a further increase in the θ (>30°), the $E_B$ position of the central peak remains the same. The HR ARXPS Cu 2p spectra at different θ do not show clear shake-up satellites (Figure 1f), which were, however, noticed during the GCIB sputtering (as explained below). The presence of shake-up satellites indicates the presence of Cu(II) species on the surface.

The shapes of the Cu $L_3M_{4,5}M_{4,5}$ spectra at all θ measured show the typical spectrum for the Cu-azole organometallic complex, with the most intensive peak located at 573 eV (Figure 1g) [40]. The shape of these spectra was similar at all θ, implying that the surface layer comprising the organometallic complex is relatively thick and the intensity of the substrate signal is relatively low. As shown below, the shape of this spectrum changes significantly when the Cu environment changes.

### 3.2. Depth Profiling Using a Gas Cluster Ion Source and Monoatomic Ar

Figure 2 shows the HR XPS spectra measured after sputtering with 4 keV $Ar_{2000}^+$ (Figure 2a), 8 keV $Ar_{150}^+$ (Figure 2b), and 4 keV $Ar^+$ (Figure 2c). In order to slowly sputter the organic surface layer, large clusters at relatively low accelerated voltage were employed first, i.e., 4 keV $Ar_{2000}^+$. This sputter beam has a low penetration power due to the low accelerated energy per atom, which is useful for slow removal of the very topmost organic species on the surface. HR XPS spectra were measured in between each sputtering cycle. The shape of the HR XPS spectra in Figure 2a,d,g,m did not change significantly after each sputtering cycle. The shape of the spectra in Figure 2d indicates that a small amount of Cu(II) species might be present in the topmost position due to the presence of a shake-up satellite [44] (at $E_B$ designated by the arrow in Figure 2). The position of the O 1s spectra at approximately 532 eV suggests that oxidized organic compounds were adsorbed at the topmost position (Figure 2j). The Cl 2p spectrum possesses a high degree of noise. However, the central peak is well-established, which confirms the presence of chloride on the surface (Figure 2p). The latter indicates that the employed sputtering source did not have enough penetration power to significantly remove the organic surface layer. The latter is also seen in Figure 3a, where the atomic concentrations of the C, N, O, Cl, and Cu elements are relatively constant during the whole sputtering procedure using 4 keV $Ar_{2000}^+$. On that basis, smaller Ar clusters with higher acceleration energies that have higher accelerated energy per atom, i.e., 8 keV $Ar_{150}^+$, should be employed. The use of the latter sputtering source induced significant changes in the XPS-excited Cu $L_3M_{4,5}M_{4,5}$ spectrum. The center of this spectrum was transferred to more negative $E_B$ (Figure 2b), indicating the gradual removal of the organometallic complex from the topmost position. The last spectrum measured (the green spectrum in the inset in Figure 2b) had the shape typical of a metallic Cu, with four clearly distinguishable peaks, with peak 2 being the most intensive. Moreover, the shake-up satellites disappeared after the first sputtering cycle using 8 keV $Ar_{150}^+$ (Figure 2e), indicating that the Cu(II) species were removed. Simultaneously, the degree of noise in the N 1s and C 1s spectra increased as the surface concentration of C and N atoms decreased (Figure 3b). Moreover, the O 1s peak was transferred to more negative $E_B$ as the excitation source reached more $Cu_2O$, i.e., the position expected for metal oxides [45–48]. The Cl 2p peak completely disappeared as chlorides were removed from the surface. Figure 2c shows that the final sputtering with monoatomic $Ar^+$ produced a further decrease in peak 3 as more $Cu_2O$ was removed from the surface. Moreover, $Ar^+$ sputtering removed most (if not all) organic material and $Cu_2O$ from the surface, as the XPS peaks in Figure 2i,l,o cannot be distinguished from the background noise after the last sputtering cycle. Furthermore, no chloride was detected during this sputtering procedure (Figure 2s). Consequently, after the last sputtering cycle with monoatomic $Ar^+$, the atomic concentrations of N, O, C, and Cl dropped to a concentration lower than the XPS detection limit.

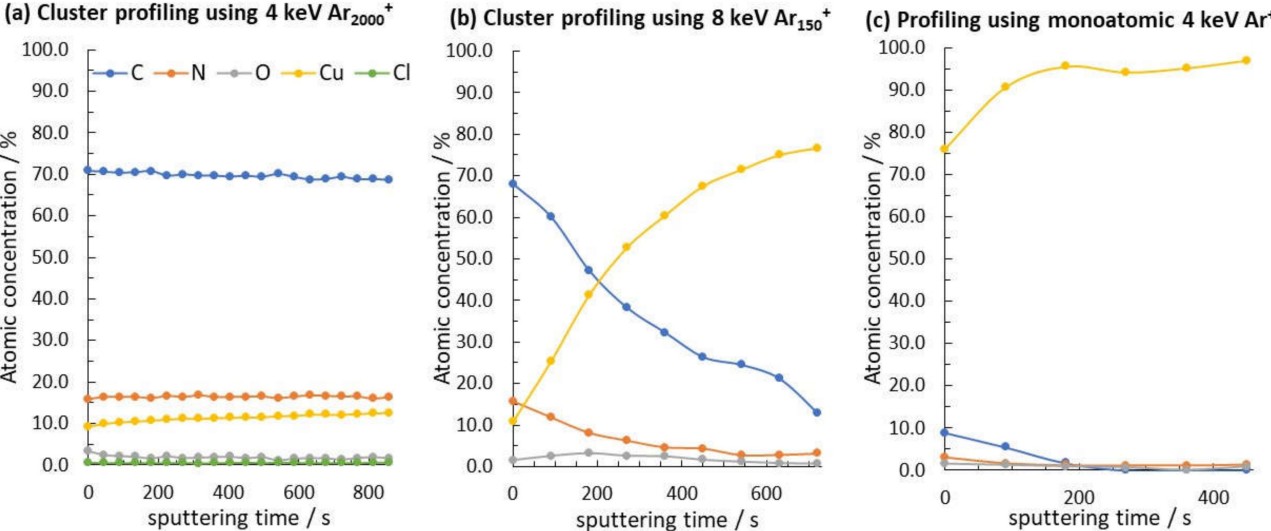

**Figure 2.** XPS analyses for the 2PhI-treated Cu: HR XPS spectra for (**a–c**) XPS-exited Auger Cu L$_3$M$_{4,5}$M$_{4,5}$, (**d–f**) Cu 2p$_{3/2}$, (**g–i**) N 1s, (**j–l**) O 1s, (**m–o**) C 1s, and (**p–s**) Cl 2p measured before and after sputtering using (**a,d,g,j,m,p**) 4 keV Ar$_{2000}$$^+$, (**b,e,h,k,n,r**) 8 keV Ar$_{150}$$^+$, and (**c, f, i, l, o, s**) monoatomic Ar$^+$ sputter beams.

**Figure 3.** The change in atomic concentration (normalized to 100.0%) with an increase in sputtering time using: (**a**) 4 keV Ar$_{2000}$$^+$, (**b**) 8 keV Ar$_{150}$$^+$, and (**c**) 4 keV Ar$^+$ sputter beams.

### 3.3. PCA in Conjunction with XPS

The XPS technique was also employed to perform imaging, i.e., to determine the surface distribution of the 2PhI molecules on the Cu surface. The analyzed spot size was 600 by 600 μm. The obtained results were further processed by performing PCA using three phases. The corresponding PCA figure and the average atomic concentration of each phase are presented in Figure 4. All three phases contain C, N, O, and Cu. Phase 1 (red) has the highest atomic concentration of C, while phase 3 has the highest atomic concentration of N. All three phases contain N, which is an indication of the presence of 2PhI, and the three phases cover the entire analyzed spot. The latter analysis will be complimentarily upgraded using ToF-SIMS imaging, which has an even higher lateral resolution and uses molecular-specific signals (as presented below).

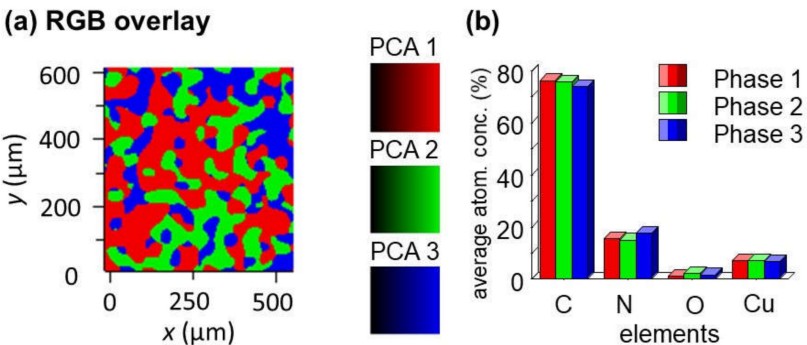

**Figure 4.** XPS imaging for 2PhI-treated Cu to obtain (**a**) the PCA-RGB image, and (**b**) the corresponding elemental average atomic concentration in each phase.

### 3.4. Molecular-Specific Analysis Using ToF-SIMS

Figure 5 shows the positive ion ToF-SIMS spectrum in the 0–290 $m/z$ range. Cu has two main isotopes at $m/z$ 62.93 and 64.93, corresponding to $^{63}Cu^+$ and $^{65}Cu^+$, respectively. These signals are shown in Figure 5. The intensive signal at $m/z$ 22.99 comes from the Na$^+$ that remained on the surface despite rinsing the 2PhI-treated Cu sample with ultrapure water. The signal for chloride was detected using XPS (as presented above, Figures 1 and 2). Therefore, some NaCl was most likely trapped inside the 2PhI surface layer. The spectrum in Figure 5 contains typical hydrocarbon-related fragments, which originate from adventitious carbonaceous species, i.e., signals at $m/z$ 15.02 for $CH_3^+$, at $m/z$ 27.02 for $C_2H_3^+$, at $m/z$ 29.04 for $C_2H_5^+$, at $m/z$ 39.03 for $C_3H_3^+$, at $m/z$ 41.04 for $C_3H_5^+$, at $m/z$ 43.05 for $C_3H_7^+$, at $m/z$ 51.02 for $C_4H_3^+$, at $m/z$ 53.04 for $C_4H_5^+$, at $m/z$ 55.05 for $C_4H_7^+$, at $m/z$ 57.07 for $C_4H_9^+$, at $m/z$ 67.05 for $C_5H_7^+$, at $m/z$ 69.07 for $C_5H_9^+$, at $m/z$ 77.04 for $C_6H_5^+$, at $m/z$ 89.04 for $C_7H_5^+$, at $m/z$ 90.05 for $C_7H_6^+$, and at $m/z$ 91.05 for $C_7H_7^+$. The signals for $C_6H_5^+$ and $C_7H_7^+$ correspond to the benzene cation and the benzyl cation (or tropylium ion), respectively [49].

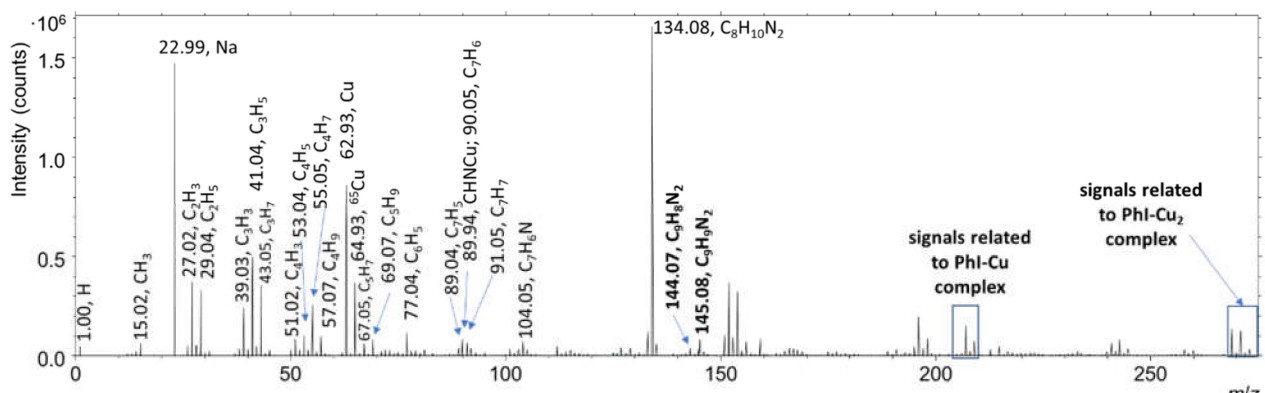

**Figure 5.** The positive ion ToF-SIMS spectrum for 2PhI-treated Cu.

The above-mentioned signals related to hydrocarbons can also arise due to the fragmentation procedure of the 2PhI molecule. Moreover, the signals at $m/z$ 104.05 for $C_7H_6N^+$ and at $m/z$ 134.08 for $C_8H_{10}N_2^+$ most likely stem from the fragmentation of 2PhI. Clear proof that the 2PhI molecule was present on the surface is the presence of signals at $m/z$ 144.07 for $C_9H_8N_2^+$ and at $m/z$ 145.08 for $C_9H_9N_2^+$ (Figure 6a). These signals correspond to the parent ion ($M^+$, i.e., $C_9H_8N_2^+$) and the parent ion with the addition of one proton (($M+H)^+$), respectively. The ToF-SIMS technique can significantly more confidently and definitively prove 2PhI adsorption compared to the above-mentioned XPS analysis, where only N-containing species and an N environment can be determined, which can leave some doubt about the presence of specific molecules on the surface. Moreover, the signal at $m/z$ 134.08 for $C_8H_{10}N_2^+$ is most likely a fragment from the 2PhI molecule.

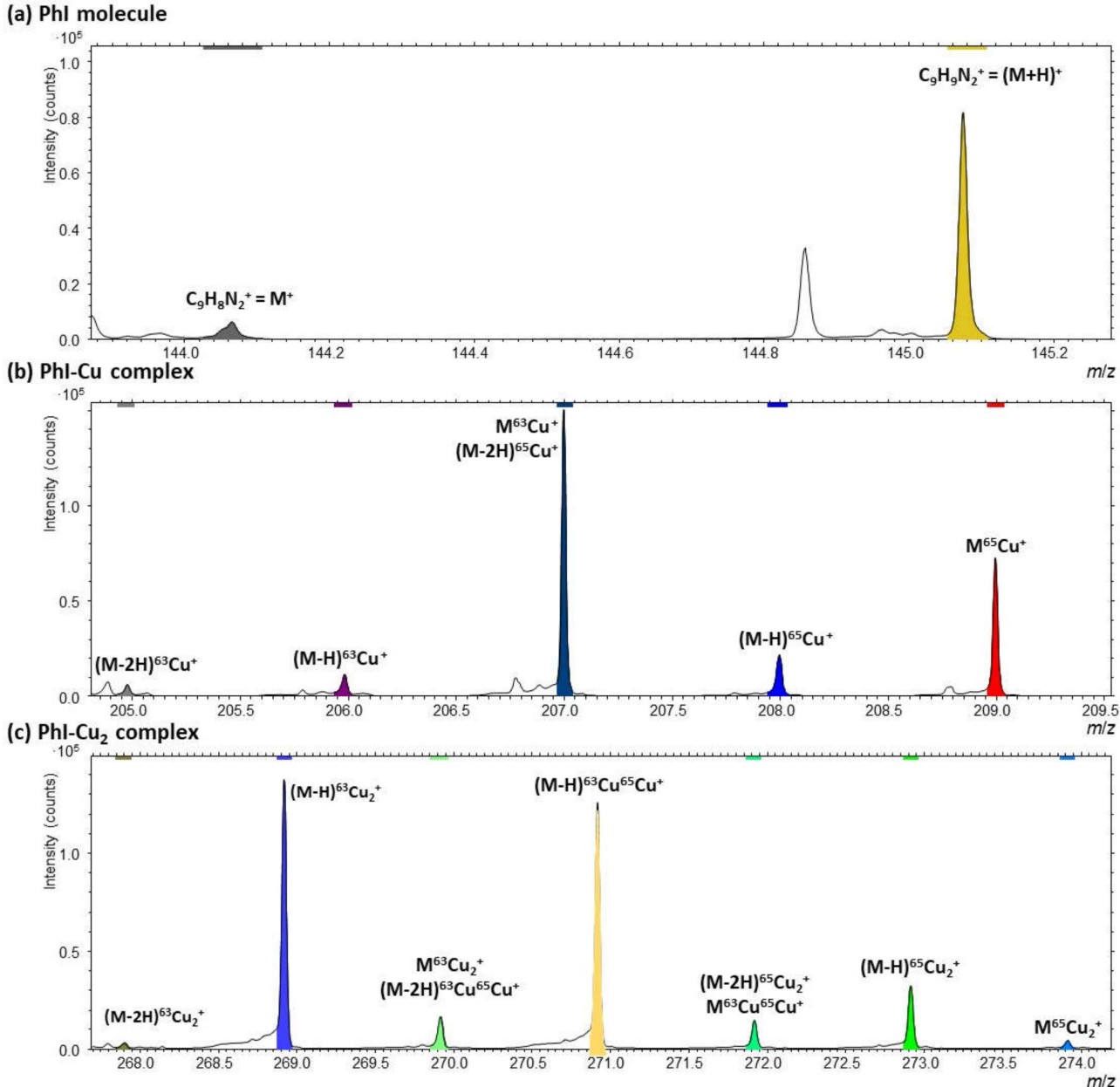

**Figure 6.** Positive ion ToF-SIMS spectra related to (**a**) the parent ion, (**b**) PhI-Cu organometallic complexes, and (**c**) PhI-Cu$_2$ organometallic complexes, which were formed on the 2PhI-treated Cu.

As already pointed out regarding the XPS measurements, the formation of organometallic complexes between copper ions and 2PhI occurred. On that basis, the signals for organometallic complexes were investigated. Figure 5 shows the signals in the $m/z$ ranges of 205–209 and 268–274 (both designated in the figure). These signals correspond to the organometallic complexes. Moreover, the signal at $m/z$ 89.94 corresponding to $CHNCu^+$ represents a fragment of the organometallic complex (Figure 5).

As copper possesses two main isotopes, more ToF-SIMS signals for different combinations representing organometallic complexes are possible, i.e., $(M–2H)^{63}Cu^+$, $(M–H)^{63}Cu^+$, $M^{63}Cu^+$, $(M–2H)^{65}Cu^+$, $(M–H)^{65}Cu^+$, and $M^{65}Cu^+$. Taking these combinations into account, a spectral interference is present for $M^{63}Cu^+$ and $(M–2H)^{65}Cu^+$, where signals for these compounds appear at $m/z$ too close to be able to separate them (Figure 6b). Moreover, organometallic complexes can be composed of two metal ions. On that basis, $(M–2H)^{63}Cu_2^+$, $(M–H)^{63}Cu_2^+$, $M^{63}Cu_2^+$, $(M–2H)^{63}Cu^{65}Cu^+$, $(M–H)^{63}Cu^{65}Cu^+$, $M^{63}Cu^{65}Cu^+$, $(M–2H)^{65}Cu_2^+$, $(M–H)^{65}Cu_2^+$, and $M^{65}Cu_2^+$ combinations are possible. By taking these compounds into account, spectral interferences are present for $M^{63}Cu_2^+$ and $(M–2H)^{63}Cu^{65}Cu^+$, and for $(M–2H)^{65}Cu_2^+$ and $M^{63}Cu^{65}Cu^+$ (Figure 6c). On the basis of the above explanation, a ToF-SIMS cooling/heating experiment was performed by employing the signals without possible spectral interferences.

### 3.5. ToF-SIMS Imaging

The sum of the signals for $^{63}Cu^+$, $^{65}Cu^+$, $^{63}Cu_2^+$, and $^{63}Cu_2^+$ was employed to characterize the Cu substrate. It is represented by the red color in Figure 7. The sum of the signals for $M^+$ and $(M+H)^+$ was employed to show the spatial distribution of the 2PhI molecules on the Cu surface (the green spots in Figure 7a). Based on Figure 7a, it can be concluded that 2PhI molecules covered most of the Cu surface. However, there were some spots that were not covered by the 2PhI molecules (the red spots in Figure 7a). Moreover, the sum of the signals that characterize the organometallic complex with one Cu ion (as designated in Figure 6b) was used. This is represented by the green spots in Figure 7b. The organometallic complex with one Cu ion additionally covers the Cu surface at the places not covered by 2PhI molecules (Figure 7a vs Figure 7b). Further analysis, by taking into account the sum of the signals representing the organometallic complex with two Cu ions (as designated in Figure 6c), showed that the green spots (in Figure 7c) match the green spots in Figure 7b. The latter suggests that organometallic complexes with one and two Cu ions were located at the same spots on the surface. The final analysis was performed by summing all the signals for 2PhI molecules (presented in Figure 6a) and all the signals for organometallic complexes (represented in Figure 6b,c) to show the spatial distribution of 2PhI-related species on the Cu surface (Figure 7d). This analysis showed that 2PhI-related species cover most of the Cu surface, which explains the high corrosion inhibition effectiveness of 2PhI for Cu in 3 wt.% NaCl solution. Therefore, the combination of 2PhI molecules and organometallic complexes act simultaneously to mitigate the corrosion of Cu.

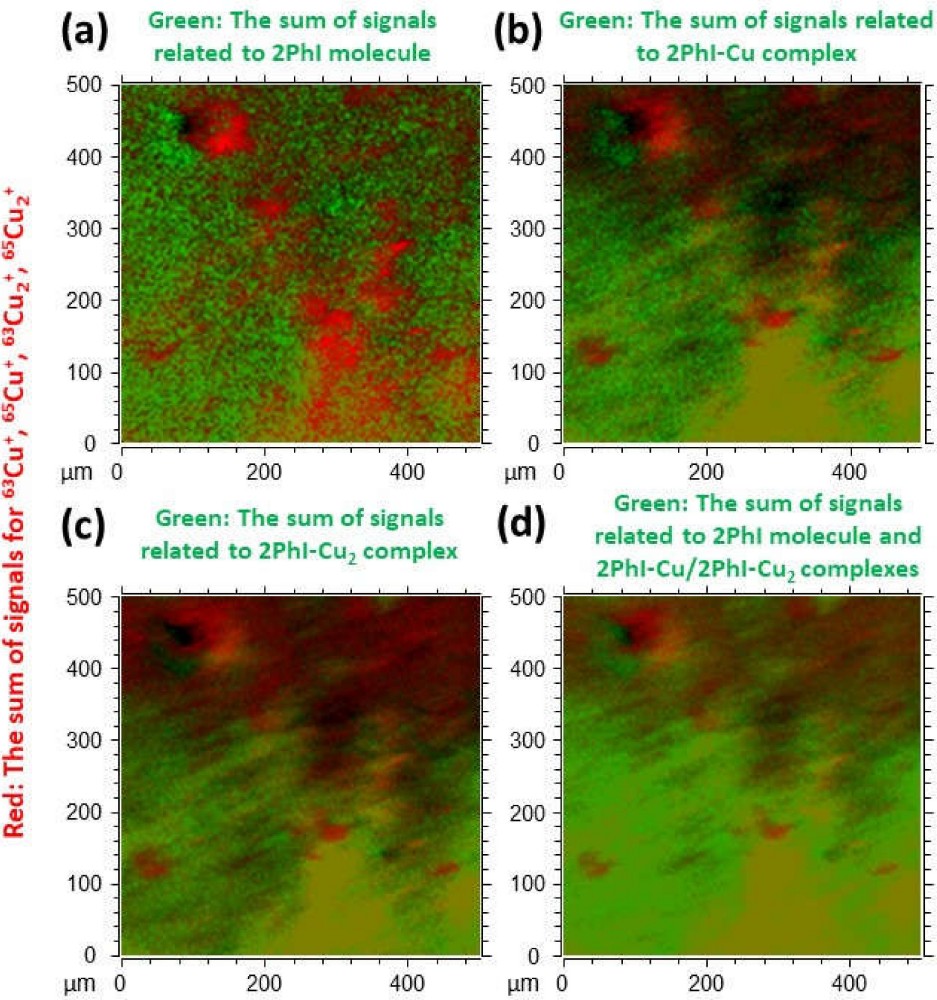

**Figure 7.** The spatial distribution of different species formed on the 2PhI-treated Cu: (**a**) 2PhI molecules, (**b**) organometallic complex with one Cu ion, and (**c**) organometallic complex with two Cu ions. subfigure (**d**) represents the sum of the signals for the 2PhI-related species in subfigures **a**–**c**. The red spots characterize the Cu substrate.

### 3.6. Temperature Persistence of the 2PhI Molecules and Organometallic Complexes on the Surface of 2PhI-Treated Cu

Temperature ($T$) frequently plays an essential role in corrosion inhibition effectiveness. A high $T$ usually results in corrosion inhibitor molecule desorption. If the corrosion inhibitor is not on the surface, the metal is prone to corrosion. The ToF-SIMS technique in association with the cooling/heating option enables the determination of the desorption temperature of corrosion inhibitors. The experiment herein started at −100 °C and the temperature of the sample was gradually elevated (0.5 °C/s) up to 500 °C. In this $T$ range, the intensity of the characteristic fragments was measured. The temperature of desorption was determined by extrapolation, as designated in Figure 8. Only the signals without spectral interferences (as explained above in Section 3.4) were taken into consideration.

The initial increase in the intensity of the signals for $(M+H)^+$ (also for $M^+$, but less intensive) in Figure 8a and $(M–H)^{63}Cu^+$ in Figure 8b in the $T$ range up to approximately 25 °C is most likely connected to the desorption of water molecules, which resulted in a higher surface concentration of $(M+H)^+$ and $(M–H)^{63}Cu^+$ species [50]. At $T$ higher than 25 °C, the intensity of all of the measured signals started to decrease due to the initial desorption process of the corrosion inhibitor molecules and their organometallic complexes. Figure 8a shows that 2PhI molecules desorb from the surface at approximately 135 °C. On the other hand, organometallic complexes desorbed from the surface at a

significantly higher temperature. The organometallic complex with one Cu ion desorbed at approximately 340 °C (Figure 8b), whereas the organometallic complex with two Cu ions desorbed at approximately 390 °C (Figure 8c).

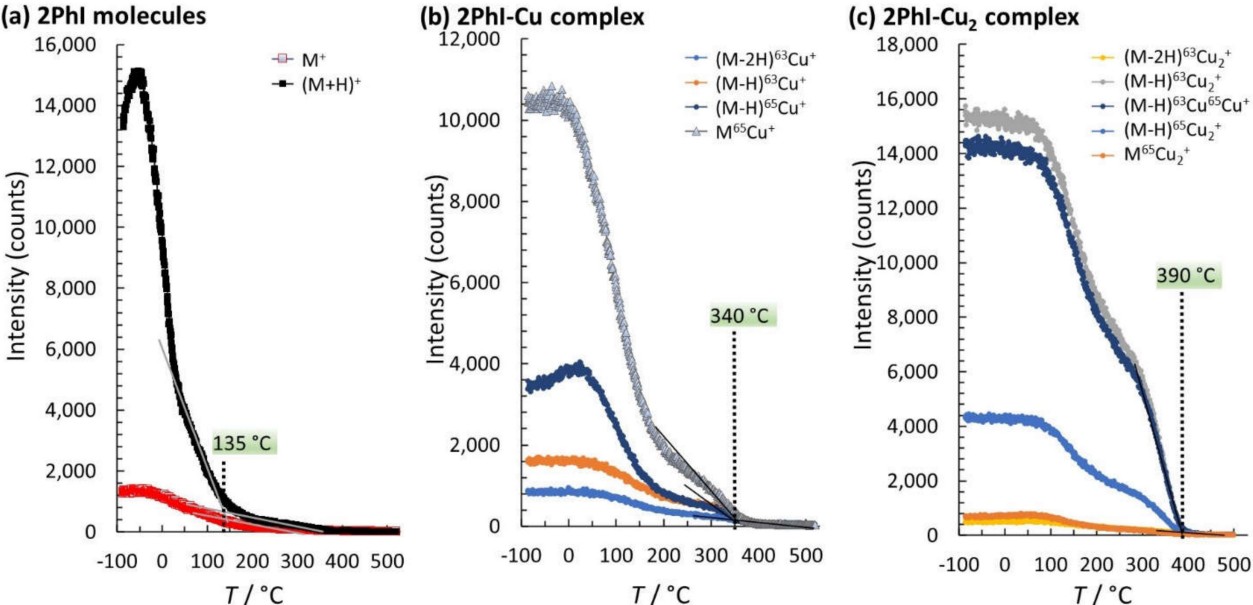

**Figure 8.** The cooling and heating experiment associated with the ToF-SIMS measurements of the signals characteristic of (**a**) 2PhI molecules, (**b**) organometallic complex with one Cu ion (2PhI-Cu), and (**c**) organometallic complex with two Cu ions (2PhI-Cu$_2$). These species were formed on 2PhI-treated Cu.

## 4. Conclusions

This work presented a surface analytical study of 2-phenylimidazole (2PhI) adsorbed on Cu surface from 3 wt.% NaCl solution. X-ray photoelectron spectroscopy and time-of-flight secondary ion mass spectrometry (ToF-SIMS) were used to investigate the surface phenomena. 2PhI is deemed to be a corrosion inhibitor for Cu in chloride-containing media. Surface analytical techniques were used to investigate the nature of surface bonding, the depth structure of the corrosion inhibitor surface layer, the 2PhI-related species distribution, and the temperature stability on the surface.

XPS analyses suggested the adsorption of the 2PhI molecules on the surface as an intense N 1s peak appeared. It was further shown that the topmost position was rich in an organometallic complex formed between 2PhI molecules and copper ions released from the substrate by the corrosion of Cu. This organometallic complex was suggested from the XPS-excited Auger Cu $L_3M_{4,5}M_{4,5}$ peak shape with an intense spectral feature at about 573 eV. The topmost position of the surface layer was also rich in oxidized carbonaceous species. In addition, the signal for chloride was also detected, indicating the presence of NaCl in the surface layer. Sputtering of the surface with a gas cluster ion beam (GCIB) resulted in the gradual removal of the surface layer species, and thus a more intense signal for Cu$_2$O and Cu. The presence of Cu(II) species cannot be excluded at the uppermost position. The feature representing Cu(II) species was removed during GCIB sputtering. The distribution of 2PhI (either as molecular 2PhI or in the organometallic complex) on the Cu surface was also analyzed by performing XPS imaging in conjunction with principal component analysis, which suggested that these species cover the entire surface. However, there were differences in the atomic concentrations for the C, N, O, and Cu elements at different locations on the surface, i.e., three phases were identified. Therefore, even though the entire surface was covered in 2PhI-related species, the surface density of these species was not uniform.

　　　　In addition, ToF-SIMS analyses confirmed molecular-specific signals for the 2PhI parent ion, providing unequivocal evidence of adsorption of the 2PhI molecules on the surface. The ToF-SIMS signals for the parent ion ($M^+$) and the parent ion with the addition of one proton were detected at $m/z$ 144.07 for $C_9H_8N_2^+$ and at $m/z$ 145.08 for $C_9H_9N_2^+$, respectively. In addition, signals were detected for organometallic complexes containing one and two copper ions and the two main copper isotopes ($^{63}Cu$ and $^{65}Cu$). These signals thus supported the XPS measurements, which suggested the formation of such complexes based on the XPS-excited Auger Cu $L_3M_{4,5}M_{4,5}$ spectra. Using the molecular-specific signals, ToF-SIMS imaging revealed the spatial distribution of 2PhI-related species on the Cu surface. By combining the ToF-SIMS signals of these species, most of the Cu surface was covered. Such coverage explains the high corrosion inhibition effect of 2PhI for Cu in 3 wt.% NaCl solution. Finally, cooling and heating experiments, in conjunction with ToF-SIMS measurements, showed that 2PhI molecules desorb first (at about 135 °C), followed by an organometallic complex with one Cu ion (at about 340 °C) and an organometallic complex with two Cu ions (at about 390 °C).

**Funding:** The author would like to acknowledge the financial support provided by the Slovenian Research Agency (Grant Number NK-0001).

**Institutional Review Board Statement:** Not applicable.

**Informed Consent Statement:** Not applicable.

**Data Availability Statement:** Data is contained within the article.

**Conflicts of Interest:** The author declares no conflict of interest.

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
