# Peer review of "2-Phenylimidazole Corrosion Inhibitor on Copper: An XPS and ToF-SIMS Surface Analytical Study"

_coatings, doi:10.3390/coatings11080966_

Round 1

Reviewer 1 Report

This manuscript reports the surface analysis of 2PhI treated Cu in NaCl solution by using a combination of XPS and TOF-SIMS techniques. Elemental and molecular specific analysis revealed spatial and depth profiles of the adsorbed species. From these results it was suggested that the adsorbed 2PhI on Cu surface was the mechanism for its corrosion inhibitor feature. The presentation is generally clear and The English is good. 

The Cu and Cu in NaCl solution without 2PhI were prepared but their results and analysis were not very clear. Lines 109-121 discussed these results but these results were never mentioned again in the subsequent sections, which makes these results less valuable. 

There seems to be some confusion regarding the Cl presence. Figure 1h (depth profile) suggest Cl is underneath some surface layer, but Figure 2 suggests Cl is at the topmost layer. This should be clarified.

Author Response

Reviewer #1

This manuscript reports the surface analysis of 2PhI treated Cu in NaCl solution by using a combination of XPS and TOF-SIMS techniques. Elemental and molecular specific analysis revealed spatial and depth profiles of the adsorbed species. From these results it was suggested that the adsorbed 2PhI on Cu surface was the mechanism for its corrosion inhibitor feature. The presentation is generally clear and The English is good.

I would like to thank the reviewer for her/his valuable comments which helped to improve the article. All issues raised have been clarified to the best of my knowledge and additional explanations have been provided as suggested. I hope that the article is now suitable for publication in Coatings.

The Cu and Cu in NaCl solution without 2PhI were prepared but their results and analysis were not very clear. Lines 109-121 discussed these results but these results were never mentioned again in the subsequent sections, which makes these results less valuable.

Answer: I would like to thank the reviewer. The analysis of bare metal (before and after immersion in chloride-containing solution) mentioned by the reviewer in lines 109-121 was summarized in the original submission. Since the bare surface of Cu (or immersed in chloride solution) has been described several times by different researchers, this has not been repeated in detail here in order to preserve the originality of the present work. There was nothing unusual about the surface, as one would not expect, and that has not been described in detail before. Since the focus of the present manuscript was on the first application of the 2-phenylimidazole corrosion inhibitor on copper, I would ask the reviewer to accept the discussion summarized as it was in the original manuscript because this article is focused to the corrosion inhibitor adsorbed on copper and not to the bare copper analyses.

There seems to be some confusion regarding the Cl presence. Figure 1h (depth profile) suggest Cl is underneath some surface layer, but Figure 2 suggests Cl is at the topmost layer. This should be clarified.

Answer: I appreciate such a close look. In this study a take-off angle was taken into account, therefore Figure 1 discussing Cl 2p spectra at 15° means that the topmost position was analyzed where Cl 2p peak was not expressed – therefore chlorides were not on the topmost position. Moreover, by increasing the take-off angle from 45° to 90°, the Cl 2p peak becomes more intense, which again suggests that the chlorides were not on the topmost position.

The depth profiling in Figure 2 was performed at 90° take-off angle, therefore the chlorides are detected immediately (before and after the sputtering procedure, Figure 2p). However, as the N 1s signal is still present during 8 keV Ar150+ sputtering (Figure 2h) and the Cl 2p peak disappears, a conclusion that chlorides were underneath the inhibitor surface layer was not appropriate. It is more likely that the chlorides were entrapped within the inhibitor surface layer.

The correction of the text was made accordingly.

Reviewer 2 Report

This is an interesting work; however, prior to proceeding to the next step, the following comments should be addressed by the authors.

  1. Language of the manuscript should be improved.
  2. Provide more in-depth discussion of related previous works.
  3. I recommend adding more discussion regarding the repeatability and reproducibility of the tests and analysis.
  4. In the “Conclusion” section, I recommend presenting more quantitative data as the main results of your research study.

Author Response

Reviewer #2

I would like to thank the reviewer for her/his valuable comments which helped to improve the article. All issues raised have been clarified to the best of my knowledge and additional explanations have been provided as suggested. I hope that the article is now suitable for publication in Coatings.

This is an interesting work; however, prior to proceeding to the next step, the following comments should be addressed by the authors.

Answer: I appreciate the positive feedback from the reviewer.

Language of the manuscript should be improved.

Answer: The English language was checked by a native English speaker before the first submission. Most of our previous papers have been checked by the same editor, I believe that the English language in this scientific paper is up to the mark.

A similar style was used previously, however, it is true that the XPS and ToF-SIMS terminology is unique.

Moreover, the other reviewer has stated that the English is good.

Provide more in-depth discussion of related previous works.

Answer: As suggested, additional text on recent studies on azoles as corrosion inhibitors for copper has been included in the Introduction.

  1. I recommend adding more discussion regarding the repeatability and reproducibility of the tests and analysis.

Answer: I appreciate the comment. The explanation about the replicate samples was additionally given in the experimental section of the revised manuscript.

The comment on repeatability and reproducibility most likely refers to concentrations determined on the same sample (different spots) or on the samples that were prepared in the same manner. The comment may be partially related to XPS, since ToF-SIMS is not a quantitative method. Moreover, XPS is also a semi-quantitative method if no internal standard is used. As mentioned below, the spectral features were reproducible when using the same XPS settings (either for ARXPS or for depth profiling). The latter was also additionally commented on in the revised manuscript. It should be noted that when using XPS, it is better to specify relative rather than absolute relations. There have been numerous unsuccessful attempts to use XPS in accreditation analysis for metals analysis, but e.g. spark-OES is still predominant. In my opinion, XPS is best suited to determine the environment (or oxidation states) of the elements, especially in conjunction with GCIB sputtering.

In the “Conclusion” section, I recommend presenting more quantitative data as the main results of your research study.

Answer: I appreciate the comment, however, this comment is not entirely clear to me. What do you mean by quantitative data?

ToF-SIMS is not a quantitative method and should not be treated as such.

If some measurable points are meant, the "numbers" regarding desorption temperature were already included in the original submission.

Regarding XPS in a “measurable” manner:

- an XPS excited Auger Cu L3M4,5M4,5 peak with an intense spectral feature at about 573 eV has been highlighted, clearly showing that the complex has been formed

- the presence of Cu(II) on the surface was discussed

- the distribution of the inhibitor surface layer on the surface based on PCA analysis was discussed, also the formation of three notable phases

If the comment was focused on the effectiveness of corrosion inhibition: this work is devoted to the study of the corrosion inhibitor mechanism and not how effective 2PhI is, for example, to determine the effectiveness of corrosion inhibition.

However, the explanation about the exact parent ion M+ can be added to the conclusions. On this basis, and as suggested by the reviewer, this information was additionally included in the Conclusions of the revised manuscript.

Round 2

Reviewer 1 Report

I am fine with the revised version